# Financial Literacy and Gender Differences: Women Choose People While Men Choose Things?

**Sigurdur Gudjonsson, Inga Minelgaite \*, Kari Kristinsson** 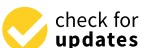 **and Sigrún Pálsdóttir**

School of Business, University of Iceland, 102 Reykjavik, Iceland
\* Correspondence: inm@hi.is

**Abstract:** According to gender personality traits, women are more interested in people, whereas men are more interested in things. The goal of this research is to see if there is a gender gap in financial literacy and if these disparities can be explained by different areas of interest. A convenience sample of nearly a thousand responses was received in quantitative research. The findings clearly show that women have lower financial literacy than men, but there is no indication that this is due to men and women's differing interests in people and things.

**Keywords:** financial literacy; personality traits; gender differences

## 1. Introduction

When comparing financial literacy between genders, it's important to remember that finance is still viewed as a masculine, male-dominated field (Bottazzi and Lusardi 2021; Jones and Merritt 2020; Boggio et al. 2014), and women have been found to have less financial knowledge, as well as less excitement for, confidence in, and willingness to learn about financial matters than men (Chen and Volpe 2002; Guðjónsson et al. 2022).

Women are generally regarded to be less confident and risk averse than their male counterparts (Croson and Gneezy 2009; Powell and Ansic 1997; Bayar et al. 2020; Yahya et al. 2020). This has been linked to their engagement, or rather lack thereof, in financial domains such as stock market participation, where men make up the majority of the players (van Rooij et al. 2011). van Rooij et al. (2011) found that women invest less in the stock market than men, a gender gap that narrows when financial literacy is taken into account (Almenberg and Dreber 2015). In the financial business, women are also paid less, are assigned worse accounts (Madden 2012), and work for smaller organizations than men (Bertrand and Hallock 2001). Only 2% of financial assets are managed by women (Sargis and Wing 2018), and globalization is likely to exacerbate the problem by requiring stockbrokers to work longer hours, which is inconvenient for family-oriented women (Blair-Loy and Jacobs 2003).

What about financial literacy among women? To emphasize the importance of financial literacy, it is worth noting that it provides people with lifelong benefits in a variety of ways, including retirement planning (Chan and Stevens 2008; Hastings et al. 2011; Bernheim and Garrett 2003; Lusardi and Mitchell 2014), mortgages (Campbell 2006; Lusardi and de Bassa Scheresberg 2013; Agarwal et al. 2009; Bertrand and Morse 2011) and being defrauded (FINRA Investor Education Foundation 2006; Blanton 2012).

General findings show that financial literacy is low for most people, regardless of gender, and this is true both among older (Lusardi and Mitchell 2011a, 2011b) and younger (Mandell 2008; Shim et al. 2010) people, as well as across countries (Mandell 2008; Shim et al. 2010; Lusardi and Mitchell 2014; van Rooij et al. 2011).

However, different demographic groupings have different levels of financial literacy. Higher-educated people outperform the uneducated (Lusardi and Mitchell 2011c; Christelis et al. 2010; Lusardi 2012), and those with high cognitive ability outperform those with low

cognitive ability (McArdle et al. 2009; Lusardi et al. 2019). Rural residents fare worse than city dwellers (Klapper and Panos 2011), and financial literacy may also be associated with different regions (Beckmann 2013; Fornero and Monticone 2011; Bumcrot et al. 2013).

As for financial literacy among women, many studies have found that women are less financially literate than men (Lusardi and Mitchell 2008; Bucher-Koenen et al. 2014; Yu et al. 2015; Lusardi et al. 2014; Atkinson and Messy 2012; Alesina et al. 2013). This has resulted in poor credit card behavior (Mottola 2013; Allgood and Walstad 2011; Allgood and Walstad 2013), and they are given worse financial credit conditions than men (Alesina et al. 2013). Similar findings have been seen in other studies. According to Zissimopoulos et al. (2015), only approximately 20% of middle-aged college-educated women could answer a basic compound interest question, whereas roughly 35% of college-educated men of the same age could.

Financial literacy is significantly linked to sociodemographic factors and family wealth, with boys from wealthy families performing particularly well (Lusardi et al. 2019). The reason for women's disadvantages could be that they learn less about finance from their parents than men do (Edwards et al. 2007; Jorgensen and Savla 2010; Newcomb and Rabow 1999), with the mother's background playing a particular role in determining the financial literacy of girls (Edwards et al. 2007; Jorgensen and Savla 2010; Newcomb and Rabow 1999; Bottazzi and Lusardi 2021).

Disparities in financial literacy between men and women may be explained by social differences between countries and ethnic groups within the same country (Nicolini et al. 2013). While women were shown to be less financially educated in rich Western countries, both women and men were found to be equally financially illiterate in developing countries (Lusardi and Mitchell 2008). Women in former West Germany outperformed women in former East Germany in terms of financial literacy (Bucher-Koenen and Lusardi 2011). Both men and women from low-income homes have inadequate financial literacy, suggesting that socioeconomic concerns such as poverty, rather than gender, are to blame (Agarwalla et al. 2015).

But what about Iceland, the gender utopia? Iceland is widely recognized as one of, if not the world's, most gender-equal countries (World Economic Forum 2020; Hausmann et al. 2011; Olafsdottir 2018; Economist 2017; Georgetown Institute for Women, Peace and Security 2017). In this paper we ask if there is gender difference in financial literacy and if that difference could be explained by different gender personality traits, i.e., women being interested in people and men in things, that has a firm supporting literature (Lippa 1998; Lippa 2010; Lippa and Dietz 2000; Schmitt et al. 2017; Costa et al. 2001). According to Baron Cohen's—empathizing-systemizing (E-S) theory, individuals are classified based on emotional intelligence, i.e., empathize quotient and systematize quotient. The theory defines emotional intelligence (e.g., empathizing) as the ability to identify and respond to the thoughts and feelings of others. However, the ability to build systems and exert control over them is defined as the systemizing quotient (Wakabayashi et al. 2006).

Women have on average more empathy and emotional intelligence than men and men are on average more systematic than women. Gender division in the labor market supports these results, as women outnumber men in social jobs, while men outnumber women in more systematic jobs such as mathematics, physics, and engineering (Wakabayashi et al. 2006). For our purposes, it is therefore possible that men have more interest in financial instruments and calculations due to their more systematic inclination. This would in turn lead to better financial literacy among men.

The sample of our research is limited to Iceland, therefore the generalizability of our results should be confined to this small country. However, it is our hope that this investigation should serve as a starting point for future research that will increase the generalizability to other countries. The structure of the paper is as follows: first, the method of the study is explained, afterwards the results are presented, and finally a conclusion is provided.

## 2. Method

In this study, a quantitative research method was applied. Three questionnaires were utilized in the survey. The respondents' gender, age range, and highestlevel of education were all queried in the background questions. The first two questions are based on a questionnaire created in 2006 for an Empathy Quotient and Systemizing Quotient study (Wakabayashi et al. 2006). There are 23 questions on the first questionnaire and 25 questions on the second. The possible responses to the first two questionnaires are offered on a six-point Likert scale: strongly disagree, disagree, neither agree nor disagree, agree, and strongly agree. Six items are included in the final questionnaire to test individuals' financial literacy. This questionnaire was adapted from the 2006 NFCS (National Financial Capability Study) and translated into Icelandic by the authors (Mottola 2013).

The survey was set up on the website www.questionpro.com in the form of an online survey. The study's population included everyone over the age of 18, and the sampling methods were convenience sample and snowball sampling. The online survey was distributed to 8363 undergraduate students at the University of Iceland as well as on Facebook. The authors planned to achieve a larger sample and better generalization value by distributing the survey in various Facebook groups and encouraging family and friends to share it further.

The data was processed in Excel and SPSS after the survey results were available. The primary background information of the subjects was plotted using descriptive statistics. Participants' responses to questionnaires one and two were graded to see if they scored higher on the emotional intelligence scale or the system intelligence scale.

The survey had a total of 1307 participants, but after the data was cleared and unsatisfactory answers were removed, the survey had 803 valid participants, and these results will be used. The study's background variables were three in number: gender (Table 1), age (Table 2), and education (Table 3), as presented here below:

**Table 1.** Gender distribution.

| Gender | Proportion | Number |
|---|---|---|
| Women | 70.11% | 563 |
| Men | 29.14% | 234 |
| Other | 0.62% | 5 |
| Don't want to say | 0.12% | 1 |

**Table 2.** Age distribution.

| Age | Proportion | Number |
|---|---|---|
| 18–19 | 47.32% | 379 |
| 30–49 | 39.32% | 314 |
| 50–69 | 12.73% | 102 |
| 70+ | 0.75% | 6 |

**Table 3.** Education distribution.

| Highest Education | Proportion | Number |
|---|---|---|
| Primary School | 5.24% | 42 |
| Highschool | 49.25% | 395 |
| University (BA/BS) | 28.05% | 225 |
| University (Graduate) | 17.46% | 140 |

Table 1 demonstrates that women make up the vast majority of participants, accounting for 70%. Men make up 29% of the participants, but because the sample is so huge, they still account for 234. Only five people, or less than 1% of the participants, identify as being of another gender.

As presented in Figure 1, the majority of participants (47%) are between the ages of 18 and 29. With 39 percent of participants, the 30–49 year old age group is the second largest. Only 1% of participants are 70 years or older, with 13% being 50–69 years old. In the background data, it can also be noted that the majority of participants (49%) have completed high school or an equivalent education. 28 percent have completed their undergraduate education and 18 percent have completed their graduate studies at a university. Merely 5% of those who took part had only finished compulsory education.

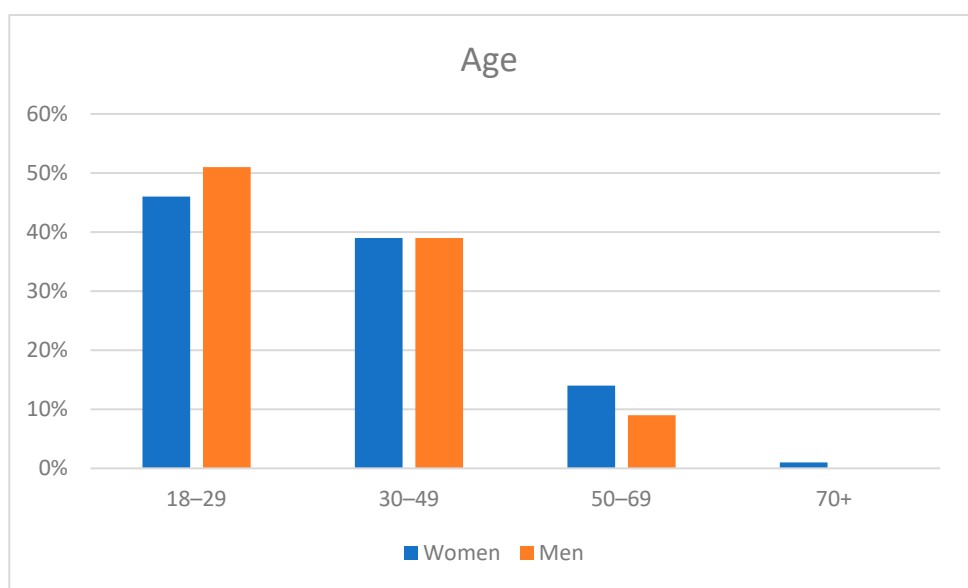

**Figure 1.** Age.

Figure 2 illustrates that there isn't much of a difference in participant education based on gender. Men are slightly more likely than women to have completed upper secondary school or a comparable education, with 51 percent of men and 48 percent of women having completed this level of schooling. Women, on the other hand, are slightly more likely than males to have completed university study at the undergraduate level, with 29 percent of women and 26 percent of men having done so.

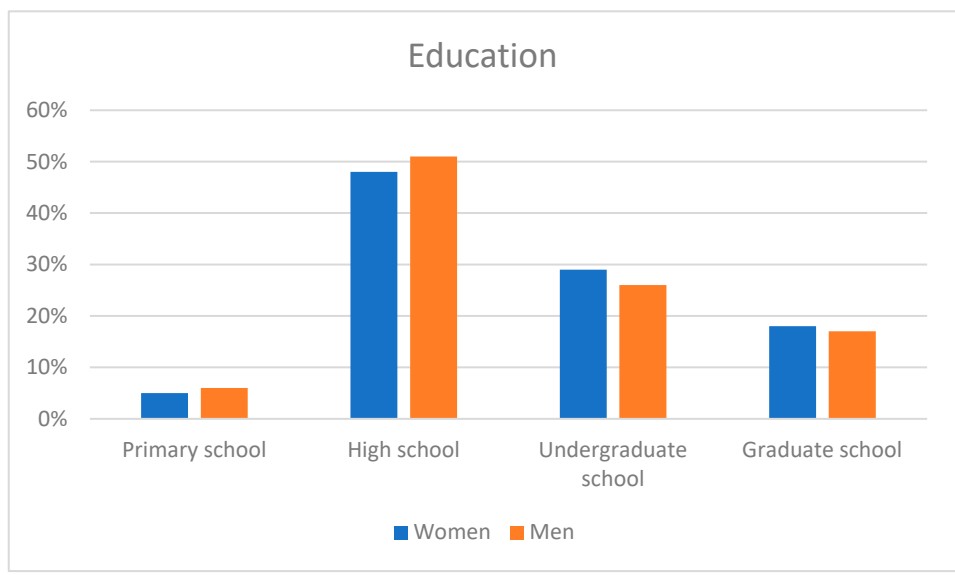

**Figure 2.** Education.

In order to examine "people vs. things", clinical psychologist Simon Baron-Cohen originally put forward the "empathizing-systemizing (E-S) theory". According to his theory, individuals are classified on the basis of emotional intelligence (PEOPLE) or the ability to empathize on the one hand and system or spatial intelligence on the other. According to this theory, empathizing is defined as the ability to identify the thoughts and feelings of others and respond to them appropriately. Systemizing (THINGS) is defined as the ability to construct a system, predict system behavior, and control it (Wakabayashi et al. 2006). The standardized questionaries were mandatory for all those who participated in the research.

## 3. Results

To begin, we examine the survey's knowledge section to determine if there is a substantial difference in financial literacy between men and women i.e., that we use to define the dependent variable. The five questions tested participants' understanding of returns, inflation, equities, bonds, and mortgages.

Question 1: "Assume you have ISK 100 in a savings account earning 2% annual interest. How much money should you have in 5 years if you don't spend it?" "More than 102 ISK" is the right response to the question. In 98 percent of the situations, men (median = 1.06 n = 232) answered correctly. In 83 percent of cases, women (median = 1.41 n = 567) answered correctly.

Question 2: "If you have a 1% annual interest rate on your savings account and inflation is 2% per year. What could you buy with the money you put into this savings account after a year?" "Less than today," is the proper response to the question. Men (median = 2.94 n = 231) properly answer the question in 86 percent of cases, while women (median = 3.12 n = 561) correctly answer the question in just 60% of situations. It's noteworthy to observe that 26% of women indicate they are unsure of the answer.

Question 3: "Is it true that buying stock in a single firm yields a higher return than investing in an equity fund?" "Wrong" is the proper response to the question. The responses of men and women to this issue are vastly different. Men (median = 2.14 n = 232) correctly answer the question 84% of the time, while women (median = 2.41 n = 558) correctly answer only 51% of the time.

Question 4: "What happens to bond prices when interest rates rise?" was the worst of all the questions for both sexes. "They diminish," is the proper response to the question. Figure 3 demonstrates that men (median = 2.56 n = 232) properly answer the question in 23% of cases, but women (median = 3.17 n = 559) only answer correctly in 13% of situations. It's noteworthy to note that while 41% of males believe bonds rise in lockstep with interest rates, only 32% of women believe this. Women say they don't know the answer 39% of the time, whereas males say the same 23% of the time.

Regarding the comparison of men's and women's answers to the fifth and final knowledge question: "Mortgages for 15 years usually carry higher monthly installments than mortgages for 30 years, but the total interest paid over the loan period is lower". The correct answer to the question is "correct". Men (median = 1.17 n = 232) 23 answer correctly in 89% of cases and women (median = 1.45) in 74% of cases.

Figure 3 illustrates the proportion of right responses for men and women for each of the five questions on the same bar chart. It demonstrates that men correctly answer all questions proportionally more often than women. It's interesting to observe how few individuals accurately answer question four.

The main result, after we have constructed our dependent variable was to run a regression with the independent variables of personal trait i.e., PEOPLE and THINGS as well as with the background variables gender, age, and education. According to the findings, women have worse financial literacy than males. After evaluating the survey's knowledge section, a regression analysis was carried out to see if there was a link between personality qualities and financial literacy, as well as to look at the impacts of gender, age, and education on financial literacy.

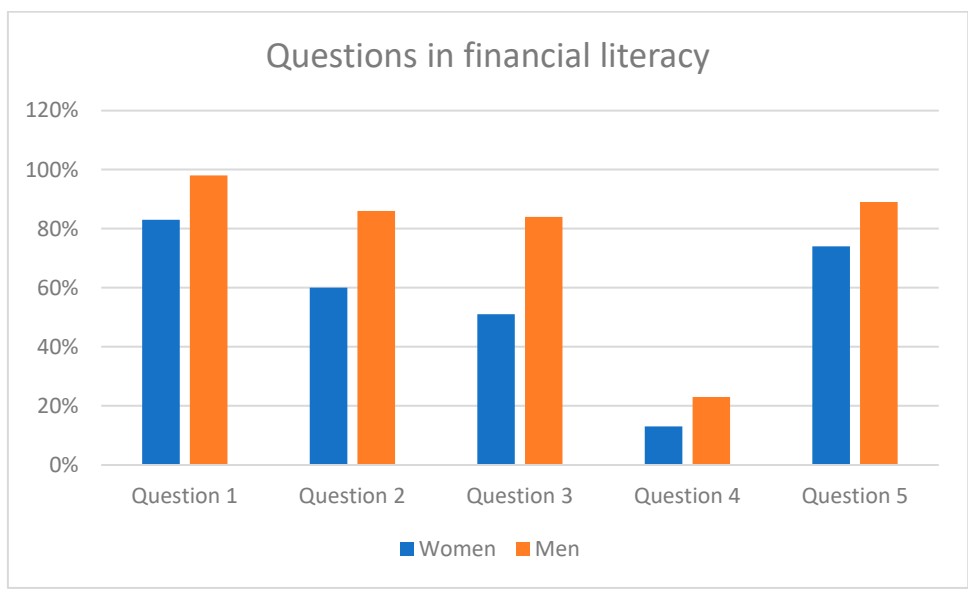

**Figure 3.** Questions in financial literacy.

The mathematical model for regression equations is:

$$Y_i = ß_0 + ß_i \, X_i + e_i.$$

The regression equation in this case is:

Financial literacy = 0.364–0.067 gender + 0.021 age + 0.048 Education + 0.019 PEOPLE + 0.029 THINGS. Men were given the number one and women the number two.

The results show that women have lower financial literacy than men, i.e., the higher the number, the worse the financial literacy. The significance threshold for the gender variable in the regression analysis is $p < 0.001$, which indicates that the results are significant. The regression study also reveals that financial literacy is affected by age, with stronger financial literacy as one gets older. The age variable has a significance level of $p < 0.13$. It indicates that the findings are significant, but age has less of an impact on financial literacy than gender. Education also affects financial literacy, but the higher the education, the better financial literacy. The significance level for the variable education is $p < 0.001$, which tells us that the results are significant, and that education has a decisive effect. Neither emotional intelligence nor systems intelligence affects financial literacy according to regression analysis.

Table 4 shows the standards used in the regression equation where the variable PEOPLE mean emotional intelligence and THINGS means systemic intelligence.

**Table 4.** Results from the regression analysis.

| Model | B | Std. Error | Beta | t | Sig. |
|---|---|---|---|---|---|
| Constant | 0.364 | 0.067 | | 5.442 | <0.001 |
| Gender | −0.67 | 0.13 | −0.188 | −5.248 | <0.001 |
| Age | 0.021 | 0.008 | 0.089 | 2.499 | 0.013 |
| Education | 0.048 | 0.007 | 0.237 | 6.624 | <0.001 |
| PEOPLE | 0.029 | 0.016 | 0.063 | 1.800 | 0.072 |
| THINGS | 0.029 | 0.016 | 0.063 | 1.800 | 0.072 |

## 4. Discussion and Implications for Further Research

The hypothesis that personality traits affect financial literacy is not supported. It was not possible to find a significant difference in financial literacy depending on whether participants had emotional intelligence or system intelligence. The authors therefore believe

that it is a worthy research project to carry out a similar study where more variables are examined that can shed light on the gender difference in financial literacy.

The results of the study show that women have less financial literacy than men. These results are not surprising but are in line with both domestic and foreign studies that have been conducted in the past (Lusardi and Mitchell 2008; Bucher-Koenen et al. 2014; Yu et al. 2015; Lusardi et al. 2014; Atkinson and Messy 2012; Alesina et al. 2013). The author finds it worrying, however, that this difference still exists even though Iceland is at the forefront when it comes to gender equality. In the World Economic Forum report for 2020, Iceland is at the top of gender equality, for the 11th year in a row (World Economic Forum 2020). It is the authors hope that this research will lead to the awakening that there is still a lot lacking when it comes to gender equality, even in Iceland. A possible explanation for the consistent gender difference in financial literacy has been suggested by Fonseca et al. (2012). They suggest that within households men are more often than women specialized in financial decisions and thereby gather more knowledge in this particular area of expertise.

Age also affects financial literacy, but with age often come certain experiences, so it is not unexpected that those who are older have better financial literacy than younger individuals. Education also affects financial literacy, as has been suggested by previous research and these results are therefore not surprising (Lusardi and Mitchell 2011c; Christelis et al. 2010; Lusardi 2012).

The authors also find it interesting to see women answer questions relatively much more often with "do not know" than men. It would be interesting to see what lies behind this. Could it simply be a matter of less understanding, or could it possibly be attributed to women's insecurity in finances (Chen and Volpe 2002)?

Like all research, this study has several limitations. First, our results are limited to Iceland, which reduces the generalizability of our findings considerably. Therefore, extending our study to other countries would be the next logical step. Secondly, our results are based on a convenience sample with a large gender skewness. A larger dataset with a more equal gender distribution could provide deeper insights into the area of inquiry. Third, the inclusion of further control variables—e.g., personality, work experience and cultural factors would improve our analysis considerably.

**Author Contributions:** Conceptualization, S.P.; methodology, K.K.; formal analysis, K.K.; investigation, S.G.; data curation, S.G.; writing—original draft preparation, S.G. and I.M.; writing—review and editing—I.M. All authors have read and agreed to the published version of the manuscript.

**Funding:** This research received no external funding.

**Informed Consent Statement:** Informed consent was obtained from all subjects involved in the study.

**Data Availability Statement:** Data can be provided upon reasonable request.

**Conflicts of Interest:** The authors declare no conflict of interest.

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
