# Peer review of "Financial Literacy and Gender Differences: Women Choose People While Men Choose Things?"

_admsci, doi:10.3390/admsci12040179_

Round 1
Reviewer 1 Report
This paper uses convenience sample data from Iceland to examine possible gender differences in financial literacy based on value preferences. The paper is generally well done. I have a few comments for revision.
1. Iceland makes for a compelling case. However, greater stress could be placed on the lack of generalizability given the convenience sample and Icelandic case study.
2. What does it mean to say that results are "unmistakable?" This charge is made in the third sentence of the Method section. Surveys and statistical data have limitations in what's asked, interpretation of questions, measurement error, etc. I'd caution against overplaying one's hand here. Please clarify.
3. The authors could stress the explanation for the gender gap in financial literacy, even in the abstract. The nonfindings are meaningful here, but what about the explanation for overall literacy differences?
4. Why are women so over-represented in the sample? Could this bias the results?
5. The results would be easier to understand if hypotheses based on theory and/or prior studies were generated and tested. Is that possible?
6. Please provide greater attention to limitations and the rectification of them through recommendations for future research.
There are some typos. The paper needs a careful proofreading. Well done, but room for improvement.
Author Response
We are very grateful to the reviewers for their knowledgeable and perceptive comments, which have led to substantial improvements to our paper. Based on their comments we have made important changes to the manuscript. Detailed responses are listed below.
Iceland makes for a compelling case. However, greater stress could be placed on the lack of generalizability given the convenience sample and Icelandic case study.
We have added two sentences to the manuscript emphasizing the lack of generalizability of our results. One sentence in the introduction and one in the discussion.
What does it mean to say that results are "unmistakable?" This charge is made in the third sentence of the Method section. Surveys and statistical data have limitations in what's asked, interpretation of questions, measurement error, etc. I'd caution against overplaying one's hand here. Please clarify.
This wording was a mistake on our part. The sentence has been removed from the manuscript.
The authors could stress the explanation for the gender gap in financial literacy, even in the abstract. The nonfindings are meaningful here, but what about the explanation for overall literacy differences?
We have added a sentence in our discussion to address this concern.
Why are women so over-represented in the sample? Could this bias the results?
We have addressed this point when discussing the limitation of our results.
The results would be easier to understand if hypotheses based on theory and/or prior studies were generated and tested. Is that possible?
|
We have added paragraphs in our introduction to further explain our theoretical framework and hypothesized results. |
Please provide greater attention to limitations and the rectification of them through recommendations for future research.
We have addressed this point when discussing the limitation of our results
There are some typos. The paper needs a careful proofreading. Well done, but room for improvement
The paper has been proofread and several typos and sentences improved.
Reviewer 2 Report
It is a very interesting topic. The authors should add literature review. Also the figures should be specified. The methodology of the research results should be prepared as well (probably some of the examinations of the results by questions and survey structure should be presented there). It will be useful to add survey in appendix.
Author Response
The authors should add literature review
We have added further literature to our paper that gives a better overview of our theoretical underpinnings.
Also the figures should be specified.
We have improved the presentation of our figures.
The methodology of the research results should be prepared as well (probably some of the examinations of the results by questions and survey structure should be presented there).
To improve the methodology, we have reworded some sentences as well as made our analysis clearer for the reader.
It will be useful to add survey in appendix
As requested, we have added our survey in appendix (in supplementary folder).
Round 2
Reviewer 1 Report
I commend the authors on a sound revision. There is an extra space above "Figure 3 illustrates..." Well done!